# Comparison of Fibrinolysis in Peripartum and Non-Pregnant Mares Using Modified Thromboelastography [note 1]

**DOI:** 10.3390/ani15131822

**Published:** 2025-06-20

**Authors:** Kira L. Epstein, Kelsey A. Hart, Ella J. Chakravarty, Steeve Giguère

**Affiliations:** Department of Large Animal Medicine, University of Georgia College of Veterinary Medicine, Athens, GA 30605, USA; khart4@uga.edu (K.A.H.); ella.chakravarty@uga.edu (E.J.C.)

**Keywords:** pregnant, mares, fibrinolysis, thromboelastography, hemorrhage, hypercoagulation

## Abstract

Hypercoagulation (increased blood clotting) and hypofibrinolysis (decreased blood-clot breakdown) occur in the peripartum period in women and are believed to protect against excessive peripartum hemorrhage. Although hypercoagulation has been identified in peripartum mares, little is known about fibrinolysis in this population. The utility of traditional coagulation testing methods that measure plasma proteins and enzymes for identifying changes in fibrinolysis is questionable. Tissue-factor (TF)-activated, tissue-plasminogen-activator (tPA)-modified global coagulation/fibrinolysis assays like thromboelastography (TEG) have been shown to reflect fibrinolytic statuses more accurately in humans and dogs but have not been evaluated in pregnant horses. The objective of this study was to characterize and compare the fibrinolytic statuses of healthy, non-pregnant mares and peripartum mares using tPA-modified TF-activated TEG. Our results demonstrated evidence of relative hypofibrinolysis in peripartum mares during late gestation and the early post-partum period compared to non-pregnant mares. These results should be interpreted cautiously given the low sample size and several unexpected findings. Further work in this area is warranted and may lead to improved clinical care in peripartum mares to minimize the risk of and optimize treatment protocols for peripartum hemorrhage in equine reproductive medicine.

## 1. Introduction

Changes in coagulation and fibrinolysis that favor clot formation and maintenance occur during pregnancy and are presumed to prevent hemorrhage during parturition. Traditional coagulation testing in women, bitches, and mares [1,2,3,4,5] and viscoelastic coagulation testing [3,5,6,7] in women have been evaluated during the peripartum period, demonstrating hypercoagulability during late gestation in women and mares [1,2,6]. In women, hypofibrinolysis occurs during late gestation [8,9], followed by a rapid increase in fibrinolysis after parturition, with normalization by 4–6 weeks post-partum [3,10]. Even with these changes in coagulation and fibrinolysis during the peripartum period, parturition-related hemorrhage remains an important cause of morbidity and mortality both in women and mares [11,12,13]. Failure to adequately suppress fibrinolysis, or even inappropriate hyperfibrinolysis, could contribute to peripartum hemorrhage [14]. Antifibrinolytics are used in both women and mares in the treatment of peripartum hemorrhage [11,12]. Despite the potential role of abnormal fibrinolysis in peripartum hemorrhage, the current understanding of fibrinolysis in the peripartum period in mares is limited.

Thromboelastography (TEG) has been used to evaluate coagulation and fibrinolysis in normal horses and in horses with gastrointestinal disease [15,16,17,18,19,20]. In horses, the ability of conventional TEG to assess fibrinolysis is limited by the minimal amount of fibrinolysis that occurs during the testing period (60 min beyond reaching the maximum amplitude [MA]) [7,15,16,18]. Similar challenges with using TEG to rapidly assess fibrinolytic potential have been identified in humans and dogs, resulting in evaluations of modified whole-blood TEG assays using tissue plasminogen activator (tPA). A standardized tPA-modified TEG technique has been used to evaluate fibrinolysis in humans [21,22]. Similar whole-blood tPA-modified TEG techniques have been used in dogs to identify abnormalities in fibrinolysis, but there is a lack of consistent concentrations of tPA and the types of activators used [23,24]. Plasma samples have been analyzed with similar modifications by a spectrophotometric global coagulation assay in humans, and it has shown similar value in evaluating fibrinolytic potential [9]. In horses, tPA-modified tissue-factor-activated- (TF-) TEG has been used to evaluate therapeutic plasma concentrations for anti-fibrinolytic drugs in pooled plasma samples from healthy horses [25]. The objective of this study was to characterize and compare the fibrinolytic potential of healthy, non-pregnant mares and peripartum mares on the same farm using tPA-modified TF-TEG in plasma to test the hypothesis that healthy pregnant mares show evidence of hypofibrinolysis during late gestation and the early post-partum period. A better understanding of fibrinolytic function in a peripartum mare could permit a more accurate diagnosis and more effective management of fibrinolytic dysfunction and periparturient hemorrhage in such a mare. A goal of this study was to determine if the tPA-modified TF-TEG was valuable for evaluating fibrinolysis in individual horse plasma samples to allow for its future use in studies on clinical cases where not all cases could be recruited from the same institution and/or performed by the same investigator.

## 2. Materials and Methods

The University of Georgia’s Institutional Animal Care and Use Committee approved all animal care, blood sample collection methods, and study protocols. A total of fifteen university-owned mares, deemed healthy based on physical examinations, were used for blood collection. The mares in the research herd were managed similarly to large breeding farms and were kept in groups on pasture supplemented with feed and hay as needed depending on pasture status, nutritional demands, and regular observations to maintain appropriate body condition. Nine of the mares were pregnant and had uneventful deliveries of clinically healthy foals and normal passage of fetal membranes. Six of the mares were non-pregnant and housed in adjacent pastures on the same farm and served as time-matched controls for the pregnant mares. The number of mares included was determined by the number of available mares eligible for each group in the research herd during the single breeding season and sampling period.

Blood was collected monthly from all mares starting in December 2012. For the pregnant mares, blood was collected until parturition and then at 1 day, 7 days, and 30 days after parturition. Once the pregnant mares had foaled, the appropriate samples taken prior to parturition were designated as 1, 2, and 3 months pre-partum. For example, if a mare foaled in April, then the banked March, February, and January samples were designated as 1, 2, and 3 months pre-partum. For the non-pregnant mares, blood was collected through April 2013. December, January, and February were used for comparison to the pregnant mares 3, 2, and 1 month pre-partum, respectively. The March (the first month of foaling) samples were used for comparison to the pregnant mares’ 1- and 7-day post-partum samples, and the April samples were used for comparison to the pregnant mares’ 30-day post-partum samples.

Blood was collected via clean jugular venipuncture and transferred to vacuum-evacuated tubes (Vacutainer, Becton Dickinson, Franklin Lakes, NJ, USA) containing 3.2% citrate, resulting in a final citrate:blood ratio of 1:9. The tubes were placed on ice and centrifuged within 30 min of collection at 1500× *g* for 10 min at 4 °C. The plasma was harvested and frozen at −80 °C prior to the batch analysis 6–10 months following collection.

The frozen plasma samples were thawed and tested within 30 min of thawing. As described by Fletcher et al. [21], reconstituted tPA (580,000 U/mL) (FACT, George King Biomedical, Overland Park, KS, USA) was added to the thawed plasma samples to result in final concentrations of 500 and 650 U/mL. These two concentrations were chosen because they resulted in moderate (clot lysis time of 23.3 min) and severe (clot lysis time of 9.9 min) results in that study when the tPA dose–response curves were performed using pooled plasma from 24 healthy horses using a 1:1800 dilution of the tissue factor (TF) solution for activation [21]. Further evaluation of these doses in the individual plasma samples was not performed. The clot lysis time was the time from when the TEG curve reached its maximum amplitude (MA), indicating maximum clot strength, until the amplitude decreased to <2 mm. Thus, it was expected that these tPA concentrations would allow for evaluation of changes in the lysis parameters 30 min after MA.

A single investigator performed the TF-activated TEG on the 500 and 650 U/mL plasma samples for each time point. The same channel of the thrombelastograph (TEG 5000, Haemonetics, Braintree, MA, USA) was used for each tPA concentration for a given horse at all time points. The machine balances and e-tests were verified prior to each test. Twenty μL of 0.2 M CaCl_2_ (Haemonetics, Braintree, MA, USA) and 10 μL of 1:100 TF (Innovin, Dade-Behring, Newark DE): in 4% bovine albumin and phosphate-buffered saline (Sigma-Aldrich, St. Louis, MO, USA) were added to the TEG reaction cups (Haemonetics, Braintree, MA, USA), followed by combining 330 μL of plasma with either 500 or 650 U/mL tPA. This resulted in a 1:3600 dilution of TF, consistent with previous TEG techniques used on normal horses and horses with gastrointestinal disease [15,16,18]. The TEG was stopped once the parameters determined at 30 min following MA were recorded.

Output of the TEG was recorded on a desktop computer using TEG analytical software (v. 4.2.95, Haemoscope, Niles, IL, USA). The TEG parameters that were analyzed were MA, CL30—(Amplitude at 30 min post MA [A30]/MA) × 100, and Ly30 (the percent decrease in area under the curve compared to MA 30 min after reaching MA). These parameters were chosen because they are routinely used. This was consistent with recommendations made in the veterinary consensus on viscoelastic testing published after the study’s design [26]. Hypofibrinolysis would be expected to result in a stronger clot formation (increased MA) and less fibrinolysis (increased CL30 and decreased Ly30) [22].

The analysis was performed with commercially available statistical software (SigmaPlot v 12.5Com, Systat Software, Inc., San Jose, CA, USA). Normality of the data and equality of the variances were assessed using the Shapiro–Wilk and Levene’s tests, respectively. Apart from the MA in the 500 U/mL tPA modified TF-TEG, the data were normally distributed. Thus, the data are expressed as least squared means (standard errors of the means). A two-way ANOVA with one factor repetition (time) was used to assess the effect of pregnancy (pregnant or non-pregnant), time (1, 2, and 3 months pre-partum and 1, 7, and 30 days post-partum), and the interactions between pregnancy and time for each TEG parameter. The data that were not normally distributed were log- or ranked-transformed. When warranted, multiple pairwise comparisons were performed using the method of Holm–Sidak. For all analyses, *p* < 0.05 was considered significant.

## 3. Results

### 3.1. General

The pregnant mares were all multiparous Quarter Horses with a mean age of 10.8 years (range of 8–18 years) at the time of foaling. Two mares foaled in March, four mares foaled in April, and three mares foaled in May. The non-pregnant mares were Thoroughbred (five), Appaloosa (one), and Saddlebred (one), with a mean age of 12.5 years (range of 9–15 years) in March of the same year. 

Due to errors in the collection or processing of the data, 6 samples (7.1%) for 7 time points (7.8%) (5 samples from 4 pregnant mares and 1 sample for the post-day 1 and 7 time-points from 1 non-pregnant mare) were missing, leaving 78 samples and 83 time points (49 pregnant samples/time points and 29 non-pregnant samples for 34 time points) that had TEG performed with 500 and 650 U/mL tPA. The addition of 500 and 650 U/mL tPA did not result in complete lysis by 30 min after MA (CL30 of 0) in any of the samples. With 500 U/mL of tPA, 22 samples (17 pregnant and 5 non-pregnant mares) had no lysis at 30 min (CL30 of 100) and 15 samples (8 pregnant and 7 non-pregnant mares) had a CL30 of 90–99.8%. Only 24 samples (14 pregnant and 10 non-pregnant mares) had a CL30 of <50%, with only 8 samples (5 pregnant and 3 non-pregnant mares) having a CL30 of <10%. With 650 U/mL of tPA, 7 samples (5 pregnant and 2 non-pregnant mares) had no lysis at 30 min (CL30 of 100) and 11 samples (8 pregnant and 3 non-pregnant mares) had a CL30 of 90–99.9%. A total of 44 samples (22 pregnant and 22 non-pregnant mares) had a CL30 of <50%, with 26 samples (13 pregnant and 13 non-pregnant mares) having a CL30 of <10%.

### 3.2. MA

In the samples that had 500 U/mL of tPA added, there was a significant interaction between pregnancy and time (*p* = 0.001). There were no significant differences over time within the pregnant and non-pregnant mares. MA was significantly greater in the pregnant mares than in the non-pregnant mares 1 month pre-partum (36.2 ± 2.2 vs. 20.8 ± 2.6 mm), 1 day post-partum (36.2 ± 2.1 vs. 15.6 ± 2.9 mm), and 7 days post-partum (33.5 ± 2.4 vs. 15.6 ± 2.9 mm) (Figure 1A).

In the samples that had 650 U/mL of tPA added, there was a significant interaction between pregnancy and time (*p* = 0.002). There were no significant differences over time within the pregnant and non-pregnant mares. MA was significantly greater in the pregnant mares than in the non-pregnant mares 3 months pre-partum (31.0 ± 2.1 vs. 21.1 ± 2.3 mm), 1 month pre-partum (34.1 ± 2.1 vs. 15.2 ± 2.3 mm), 1 day post-partum (32.6 ± 1.9 vs. 13.9 ± 2.6 mm), and 7 days post-partum (28.3 ± 2.2 vs. 13.9 ± 2.6 mm) (Figure 2A).

### 3.3. CL30

In the samples that had 500 U/mL of tPA added, there was a significant interaction between pregnancy and time (*p* = 0.001). There were no significant differences over time within the pregnant mares. In the non-pregnant mares, CL30 was greater (less fibrinolysis) 2 months pre-partum (88.2 ± 11.6%) and 1 month post-partum (88.3 ± 11.6%) than 1 day post-partum (28.2 ± 12.9%) and 7 days post-partum (28.2 ± 12.9%) (Figure 1B). CL30 was greater (less fibrinolysis) in the pregnant mares than in the non-pregnant mares 1 day post-partum (79.0 ± 9.4 vs. 28.2 ± 12.9%) and 7 days post-partum (69.2 ± 10.9 vs. 28.2± 123.9%) (Figure 1B).

In the samples that had 650 U/mL of tPA added, there was a significant interaction between pregnancy and time (*p* = 0.008). There were no significant differences over time within the pregnant and non-pregnant mares. CL30 was significantly greater (less fibrinolysis) in the pregnant mares than in the non-pregnant mares 1 month pre-partum (64.6 ± 11.1 vs. 20.9 ± 12.7%) and 1 day post-partum (62.5 ± 10.3 vs. 9.7 ± 14.2%) (Figure 2B).

### 3.4. LY30

In the samples that had 500 U/mL of tPA added, there was a significant interaction between pregnancy and time (*p* = 0.001). There were no significant differences over time within the pregnant mares. In the non-pregnant mares, Ly30 was significantly lower (less fibrinolysis) 2 months pre-partum (10.0 ± 10.1%) and 1 month post-partum (35.2 ± 10.1%) than 1 day post-partum (63.3 ± 11.3%) and 7 days post-partum (63.3 ± 11.3%) (Figure 1C). Ly30 was lower (less fibrinolysis) in the pregnant mares than in the non-pregnant mares 1 month pre-partum (3.2 ± 8.9 vs. 35.2 ± 10.1%), 1 day post-partum (16.9 ± 8.3 vs. 63.3 ± 11.3%), and 7 days post-partum (19.9 ± 9.6 vs. 63.3 ± 11.3%) (Figure 1C).

In the samples that had 650 U/mL of tPA added, there was a significant interaction between pregnancy and time (*p* = 0.004). There were no significant differences over time within the pregnant and non-pregnant mares. Ly30 was significantly lower (less fibrinolysis) in the pregnant mares than in the non-pregnant mares 1 month pre-partum (18.9 ± 9.9 vs. 72.5 ± 11.3%) and 1 day post-partum (26.2 ± 9.3 vs. 80.8 ± 12.7%) (Figure 2C).

## 4. Discussion

Using TF-activated TEG with tPA-induced hyperfibrinolysis, the present study documented hypofibrinolysis in pregnant mares compared to non-pregnant mares, as evidenced by increased MA and CL30 and decreased Ly30. These changes were most frequently detected in the samples closest to the parturition period, at 1 month pre-partum as well as at 1 and 7 days post-partum. By 30 days post-partum, fibrinolysis in the post-partum mares was comparable to that of the non-pregnant mares. These findings provide preliminary evidence supportive of our hypothesis that healthy pregnant mares have evidence of hypofibrinolysis during late gestation and the early post-partum period, as is seen in women [3,6,8,9]. A larger study performed after further optimization of the tPA-modified TF-TEG is warranted and would be needed to confirm our hypothesis.

There were unexpected differences within the non-pregnant mares over time in the CL30 and Ly30 using modified TEG with 500 U/mL tPA. The reason for the increased amount of fibrinolysis in the March sample (1 and 7 days) was not explained by a single outlier. It is possible that these differences were related to differences over time in the non-pregnant group and/or in comparison to the pregnant group. In humans, changes in fibrinolysis associated with a variety of temporal and non-temporal factors including season [27,28,29], sex [30,31], age [31], race [31], reproductive cycle [30,32,33], and exercise [27,34,35,36] have been identified. Not all of these potential factors could be matched between the pregnant and non-pregnant mares. We chose to use time-matched samples from non-pregnant mares as controls. We felt it was important to use mares because of the reported differences in fibrinolysis in women related to sex [30,31] and reproductive cycle [30,32,33]. It would have been ideal to utilize the same mares prior to their next breeding cycles as their own non-pregnant controls. However, given the length of gestation in mares and the routine protocol in this herd of rebreeding all mares on their first or second estrus cycle following foaling, it was not possible to sample the same broodmares while not pregnant or within the timeframe of potential immediate post-partum changes in fibrinolysis. We felt that it was important to use time-matched comparisons due to the reported differences in humans related to season [27,28,29]. Due to the composition of our research herd of mares, we were not able to match the breeds of the mares. Differences in coagulation and fibrinolysis may occur with differences in breed as they do in humans related to race [31]. However, there have not been any studies that have investigated breed-differences in horses, and the mares in the control group represented three breeds in a diverse population of horses under the same management practices and environment. 

It is important to note that these changes over time in the non-pregnant mare group made interpretation of the differences in these parameters between the pregnant and non-pregnant mares at 1 and 7 days post-partum more difficult. However, since the changes in CL30 and Ly30 over time in the non-pregnant mares were not seen in the 650 U/mL tPA-modified TEG and the differences between the pregnant and non-pregnant mares at 1 day post-partum (CL30 and Ly30) and 7 days post-partum (Ly30) were also observed, it seems likely that the relative hypofibrinolysis in the pregnant mares was not solely the result of the increased fibrinolysis in the non-pregnant mares at that time point. Additionally, while no statistical differences in parameters over time were detected in the pregnant mares, the means of the MA and CL30 were higher and the Ly30 was lower in the pregnant mares at 1 month pre-partum and 1 day post-partum compared to the other time-points, particularly in the 500 U/mL tPA-modified TEG samples.

Another unexpected finding was that neither the 500 or the 650 U/mL tPA induced as much fibrinolysis as anticipated [21]. As noted, these doses were chosen because they resulted in 100% lysis in 23.3 and 9.9 min, respectively, in order to assure adequate fibrinolysis to evaluate within 30 min of reaching MA. However, in this population, no samples had a CL30 of 0% and only ~10% of the 500 U/mL tPA samples and ~33% of the 650 U/mL had a CL30 of <10%. The pooled plasma sample used for establishing the amount of tPA used for this study was from a mixed sex population of horses. Women are hypofibrinolytic compared to men [30,31]. If mares, overall, are also hypofibrinolytic compared to geldings and stallions, then the amount of tPA required to induce fibrinolysis would be expected to be higher in mares and may have accounted for the decreased fibrinolysis overall observed in this study. To the authors’ knowledge, differences in coagulation and fibrinolysis parameters for TEG in horses related to sex have not been fully investigated or documented.

For this study, we chose to use frozen plasma samples rather than whole-blood samples based on the successful use of pooled plasma in a previous study on horses [21] and the previously noted value of plasma in a similarly modified using a spectrophotometric global coagulation assay used on humans to evaluate fibrinolytic potential [9]. Standardization of sampling, sample handling, and testing techniques for coagulation assessment in veterinary medicine is an area of much debate and ongoing research [37]. Given the limited availability of TEG, the requirements for the rapid analysis of whole-blood samples, and the established variability between users [15], the ability to perform this technique on frozen plasma was important for consistent sample analysis in this study and will also be important in future studies. However, obviously, using plasma for this assessment precluded assessment of the cellular components impacting coagulation and fibrinolysis in this study.

This study had several other limitations, including incomplete assay optimization, a lack of additional coagulation and fibrinolysis testing, and a small sample size. Firstly, we did not perform optimization testing for the TF or tPA concentrations. The use of an activator like TF is important for speeding the test up to provide rapid results and to reduce result variability [15]. However, excessive activator concentrations have the potential to mask more subtle alterations in patients, particularly for the coagulation parameters (R, K, Ang, and MA). For an assay designed to evaluate fibrinolysis, increased activation for rapid, consistent results may be prioritized over decreased sensitivity for subtle changes. The tPA concentrations need to be optimized to induce an adequate, consistent amount of fibrinolysis while avoiding resulting in so much fibrinolysis that clots are broken down so quickly that TEG tracing becomes flat and parameters cannot be measured. The technique for tPA-modified TF-TEG evaluation of fibrinolysis developed in humans [21,22] involved the optimization of TF concentrations and tPA concentrations. One technique used in dogs [23] involved the use of different activators (not optimized) and optimization of tPA concentrations. However, other investigators using tPA-modified TF-TEG in dogs did not appear to have performed any optimization and utilized significantly different tPA and TF concentrations from the other study [24]. We did not perform these steps and instead based our TF and tPA concentrations on previous work on horses that established effective concentrations of TF [15,19] and tPA [21] for this method. In the study herein, we did note a high level of inter-horse variability and an inadequate fibrinolytic response in many of the horses with this methodology, which indicated a need for optimization of the TF and tPA concentrations to be performed. The assay used here used twice the concentration of TF than previous studies [15,18,19] used on horses, but increasing the concentration further may improve the speed and consistency of the assay. Given the inadequate fibrinolytic results with the tPA concentrations, higher concentrations of tPA should be tested. Importantly, horses appear to have a high level of inter-horse variability. The variability reported here was similar to that documented in native (not activated) TEG parameters in normal horses [15,17,19]. While optimization will improve the assay, there may be limitations to producing consistent results in horses even with optimization. 

Second, due to study funding constraints, we were unable to perform additional plasma-based coagulation and fibrinolysis testing. The inclusion of tests with established reference ranges would have confirmed normal coagulation and fibrinolysis and provided a mechanism for direct comparisons with the literature. However, plasma-based testing of individual fibrinolysis-related proteins has been shown to have limited value in predicting in vivo fibrinolysis in humans [38], and there were no significant correlations between fibrinogen, D-dimers, or platelet counts and Ly30 in one study that investigated tPA-modified TF-TEG in dogs [24]. Fibrinogen concentrations were also not associated with TF-TEG MA in one study on normal horses [15]. Thus, it is unlikely that the inclusion of these parameters would have substantially altered our study findings with regards to the evaluation of fibrinolysis in periparturient mares.

Finally, for this preliminary study, we were only able to sample six to nine animals per group with the animals and funding available. While a minimum of 20 subjects is generally recommended for establishing a normal range, lower numbers of horses are often used and considered acceptable for a comparison of two groups. Although some significant differences between the groups that were consistent with hypofibrinolysis during late gestation and the immediate post-partum period were detected, the current study may be underpowered for fully characterizing all of the differences in fibrinolysis that could be present in pregnant mares.

## 5. Conclusions

Overall, we detected evidence of hypofibrinolysis in peripartum mares using tPA-modified TF-TEG at 1 month prepartum and during the immediate post-partum period. However, a small sample size, the unexpected changes in fibrinolysis identified in the non-pregnant mares over time, and the finding that the tPA-modified TF-TEG did not perform as predicted consistently made it important to view these findings as preliminary and interpret them cautiously. It is necessary to better optimize the tPA-modified TF-TEG technique and perform a study with a larger cohort to confirm the results. Further investigations on fibrinolytic potential in mares with peripartum hemorrhage and horses with other conditions predisposing to hyper- and/or hypo-fibrinolysis are warranted. 

## Figures and Tables

**Figure 1 animals-15-01822-f001:**
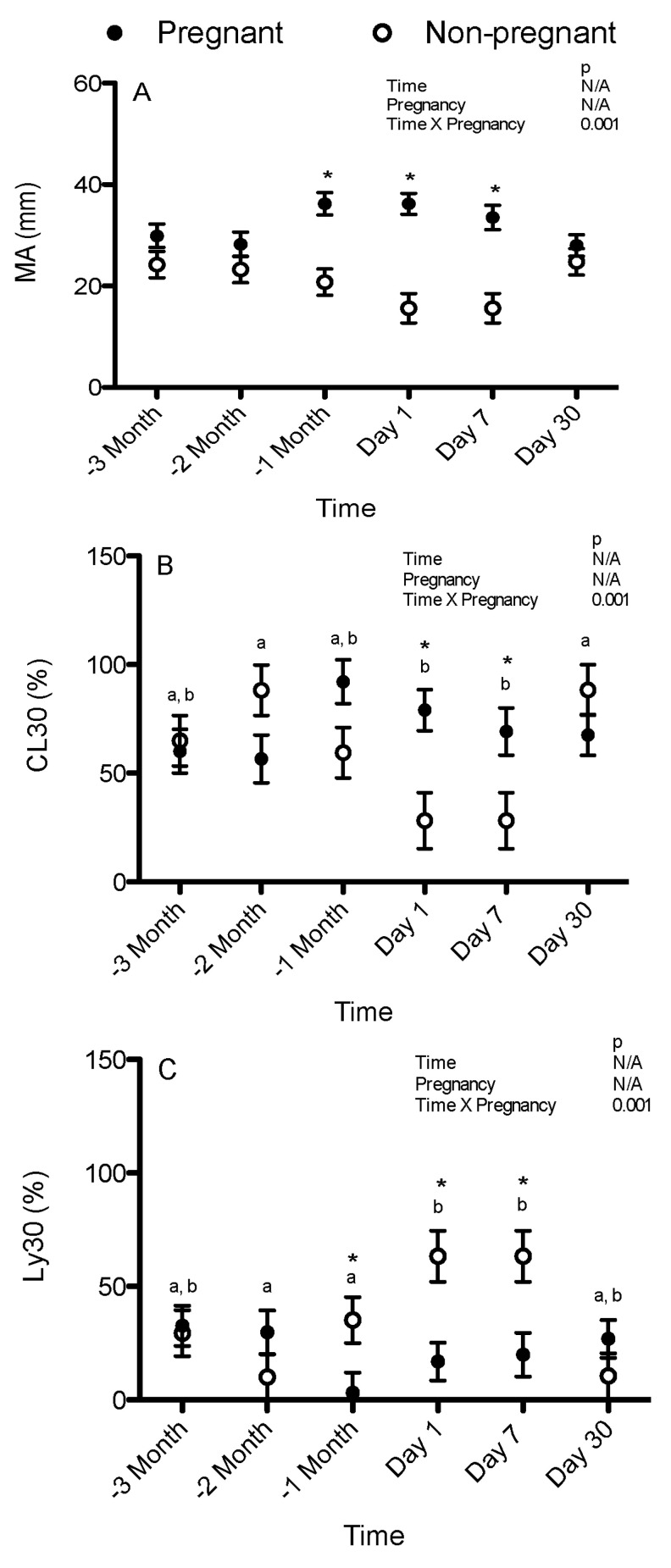
Maximum amplitude/peak for clot strength (MA) (**A**), (A30/MA) × 100 (CL30) (**B**), and the percent decrease in the area under the curve compared to the MA 30 min after reaching MA (Ly30) (**C**) in the TEG performed with 500 U/mL tissue plasminogen activator added for the pregnant and non-pregnant mares over time (means and standard errors of the means). −3 month = 3 months pre-partum, −2 month = 2 months pre-partum, −1 month = 1 month pre-partum, Day 1 = 1 day post-partum, Day 7 = 7 days post-partum, day 30 = 30 days post-partum. The different lowercase letters indicate significant differences over time in the non-pregnant mares (the time-points labeled with only a are different from those labeled with only b and the time-points labeled a, b are not different from any other time-points), and * indicates that a pregnant mare was significantly different from a non-pregnant mare. N/A indicates not applicable due to interactions identified between time and pregnancy.

**Figure 2 animals-15-01822-f002:**
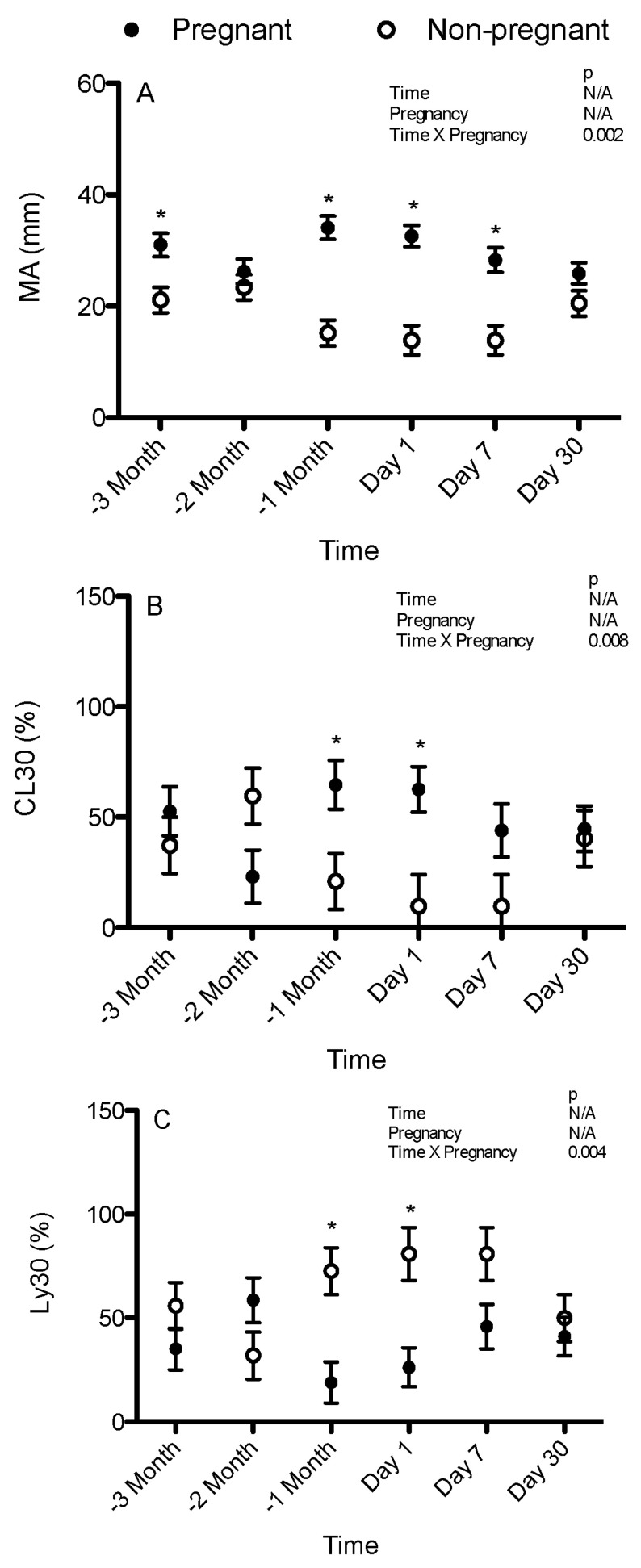
Maximum amplitude/peak for clot strength (MA) (**A**), (A30/MA) × 100 (CL30) (**B**), and the percent decrease in the area under the curve compared to the MA 30 min after reaching MA (Ly30) (**C**) in the TEG performed with 650 U/mL tissue plasminogen activator added for the pregnant and non-pregnant mares over time (means and standard errors of the means). −3 month = 3 months pre-partum, −2 month = 2 months pre-partum, −1 month = 1 month pre-partum, Day 1 = 1 day post-partum, Day 7 = 7 days post-partum, day 30 = 30 days post-partum. * indicates that a pregnant mare was significantly different from a non-pregnant mare. N/A indicates not applicable due to interactions identified between time and pregnancy.

## Data Availability

The data that support the findings of this study are now openly available on Mendeley Data, https://data.mendeley.com/datasets/mgw4g49t24/1 (accessed on 17 June 2025).

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
