# Peer review of "Comparison of Fibrinolysis in Peripartum and Non-Pregnant Mares Using Modified Thromboelastography†"

_animals, 2025, doi:10.3390/ani15131822_

Round 1

Reviewer 1 Report

Comments and Suggestions for Authors

The study examines the fibrinolytic potential in peripartum mares compared to healthy non-pregnant mares, addressing a notable gap in current equine reproductive health research. The primary objective is to elucidate the phenomena of hypofibrinolysis and hypercoagulation during pregnancy, which are crucial for minimising haemorrhage during parturition, especially in women.

The results of this study indicate some hypofibrinolytic status in pregnant mares compared to non-pregnant ones, which may align with the physiological adaptations necessary for successful delivery and then uterine involution. However, there are some limitations, such as a small sample size and unexpected changes in fibrinolytic activity observed in the non-pregnant mares over time. The authors have rightly urged caution in interpreting their findings, highlighting the need for further investigation on a larger scale.

Changes that I would suggest for the authors to implement in their paper are:

-changing from standard error to standard deviation (standard error is challenging to interpret for an average reader)

-also it would be beneficial for open science if the authors published their raw data in one of the public repositories

In summary, this study represents a valuable contribution to the understanding of coagulation dynamics in peripartum mares while also indicating the need for additional research to more fully define the implications of these findings in clinical settings.

Author Response

Comment 1:  The study examines the fibrinolytic potential in peripartum mares compared to healthy non-pregnant mares, addressing a notable gap in current equine reproductive health research. The primary objective is to elucidate the phenomena of hypofibrinolysis and hypercoagulation during pregnancy, which are crucial for minimising haemorrhage during parturition, especially in women.

The results of this study indicate some hypofibrinolytic status in pregnant mares compared to non-pregnant ones, which may align with the physiological adaptations necessary for successful delivery and then uterine involution. However, there are some limitations, such as a small sample size and unexpected changes in fibrinolytic activity observed in the non-pregnant mares over time. The authors have rightly urged caution in interpreting their findings, highlighting the need for further investigation on a larger scale. 

In summary, this study represents a valuable contribution to the understanding of coagulation dynamics in peripartum mares while also indicating the need for additional research to more fully define the implications of these findings in clinical settings.

Response 1:  The authors thank you for your time and thoughtful review.  We have addressed your concerns to our best ability and believe the manuscript has been improved based on your suggestions.  We recognize the limitations of the study design and have added to our previous discussion of these issues including sample size and testing methodology to emphasize the need for cautious interpretation and further investigation.  Additionally, we have tempered our conclusions in the abstract and end of the manuscript.  New text is highlighted in yellow and text that has been moved to improve flow or clarity is highlighted in blue.

Comment 2:  Changes that I would suggest for the authors to implement in their paper are:

-changing from standard error to standard deviation (standard error is challenging to interpret for an average reader)

Response 2:  Thank you for this comment. We agree that many readers are familiar with using the standard deviation (SD) to provide a description of the spread of data around the mean in the sample population. However, we feel that in this study, the standard error of the mean (SEM) allows us to better reflect the precision of the sample mean as an estimate of the population mean in the presentation of our data. The SEM accounts for both the sample variability and the sample size, providing a measure of how much the sample mean is likely to differ from the true population mean. Therefore, given the small sample size in this study (which we have further addressed as a study limitation in the discussion) and the wide individual variation in some parameters in this population, we do believe that the SEM is more appropriate for conveying the reliability of the mean estimates provided in our analysis and have elected to retain that parameter in our data presentation. We were not able to find specific guidance for preference of the SD vs. SEM in the Instructions for Authors, but upon review of recent publications in Animals did note that many manuscripts have represented similar data sets with the mean +/- SEM (ex: Gañán et al, Animals 2025, 15(12), 1680; https://doi.org/10.3390/ani15121680; Li et al Animals 2025, 15(11), 1673; https://doi.org/10.3390/ani15111673).  However, if the reviewer or editor feel it is desirable, we would be happy to also provide the mean +/- standard deviations for the measured parameters as a supplementary figure.

Comment 3:  Changes that I would suggest for the authors to implement in their paper are:

-also it would be beneficial for open science if the authors published their raw data in one of the public repositories

Response 3:  The authors agree and have uploaded the raw data to Mendeley Data (http://doi: 10.17632/mgw4g49t24.1).  This information has been added to the manuscript (page 11, lines 389-90).

Reviewer 2 Report

Comments and Suggestions for Authors

To the authors: Thank you for this interesting manuscript! Overall, this manuscript provides valuable preliminary information into the investigation of fibrinolysis in both health and pregnant mares and opens some interesting down the road questions. I think there are a few holes in the manuscript due to underpowered numbers and lack of complete validation, but I also understand financial constraints/case constrains that we face in equine medicine.  I just have a few points that I think would improve the overall clarity of this manuscript.

Abstract: OK

Introduction:

Line 72: From the introduction is sounds like the method you used has not been previously validated in the horse. Reading through the article it appears due to financial constraints this assay was not completely validated.  Unless there is an equine reference to add I think a reasonable goal would be preliminary investigation into the new assays use in equine patients.

Materials and Methods:

Line 88-96: It may be useful to add a schematic of this. I had to write it out the paragraph to organize when each population was being sampled.

Line 102: From the introduction it sounds like you used a previously unvalidated method. Did you take any validation steps for the assay?

Line 130: As one of your limitations is the study is under powered it may be useful to add your power calculation here.

Results: OK

Figures:

It could just be how the upload occurred, but on my screen these figures are very small, and the text is not clear.

For your lettering is the a,b different non-pregnant mares? It is slightly confusing the way this is presented.

Discussion:

Line 230: While I do think your hypothesis is supported, in your limitations you list that the assay was not optimized completely, and your study is underpowered. I wonder if it would be fair to say in these initial findings and that more work would be needed to fully determine if there is a difference as seen in humans.

Line 232: Do you think breed variation could have come into play for this? Your pregnant mare group was all AQHA where the non-group was very mixed. I think you do list this in your limitations but it may be useful to mention this here as well.

Line 250: Is it possible his was just not a high enough dose? I think you briefly mention this in your limitations but may want to expand on this a little bit more.

Author Response

Comment 1:  To the authors: Thank you for this interesting manuscript! Overall, this manuscript provides valuable preliminary information into the investigation of fibrinolysis in both health and pregnant mares and opens some interesting down the road questions. I think there are a few holes in the manuscript due to underpowered numbers and lack of complete validation, but I also understand financial constraints/case constrains that we face in equine medicine.  I just have a few points that I think would improve the overall clarity of this manuscript.

Response 1:  The authors thank you for your time and thoughtful review.  We have addressed your concerns to our best ability and believe the manuscript has been improved based on your suggestions.  We recognize the limitations of the study design and have added to our previous discussion of these issues including sample size and testing methodology to emphasize the need for cautious interpretation and further investigation.  Additionally, we have tempered our conclusions in the abstract and end of the manuscript.  New text is highlighted in yellow and text that has been moved to improve flow or clarity is highlighted in blue.

Abstract: OK

Introduction:

Comment 2:  Line 72: From the introduction is sounds like the method you used has not been previously validated in the horse. Reading through the article it appears due to financial constraints this assay was not completely validated.  Unless there is an equine reference to add I think a reasonable goal would be preliminary investigation into the new assays use in equine patients.

Response 2:  Although not validated in individual horses, the technique used in this study was taken directly from the study by Fletcher et al (reference 21).  Specifically, TF was added to the same concentration and the doses of tPA used were selected based on tPA dose response curves created using pooled plasma from 24 horses (see pg 3, line 121-125).  Further discussion of the limitations of the methodology validation have been added (see pg 9-10 line 323-351).    We have also added a goal to investigate the assay in individual horse’s plasma to allow for future studies (pg 2, lines 84-88).

Materials and Methods:

Comment 3:  Line 88-96: It may be useful to add a schematic of this. I had to write it out the paragraph to organize when each population was being sampled.

Response 3:  The authors agree that as written sample timing was not clear.  We were unable to design a clear schematic that worked for both pregnant and non-pregnant mares.  Instead, we have edited the paragraph in an attempt to clarify including an example for pregnant mares (see pg 3, line 102-112).

Comment 4:  Line 102: From the introduction it sounds like you used a previously unvalidated method. Did you take any validation steps for the assay?

Response 4:  As noted in response 2, we chose the tPA doses based on dose response curves completed in pooled equine plamsa and did not further evaluate the tPA doses in individual horses plasma.  We recognize this is a limitation of the study and have expanded our discussion of this limitation (see pg 9-10, line 323-351).  Additionally, we have added to this paragraph to clarify how the doses were chosen and that no further evaluation/validation was performed in individual horse plasma samples (see pg 2, line 121-125).

Comment 5:  Line 130: As one of your limitations is the study is under powered it may be useful to add your power calculation here.

Response 5:  The authors recognize this limitation of the study.  For this study, an a priori power calculation was not performed.  Instead, we sampled all available mares eligible for the groups in our research herd.  During that breeding season and sampling period we had 9 pregnant mares that had uneventful deliveries of clinically healthy foals and normal passage of fetal membranes, and six non-pregnant mares that were not being utilized for other studies that would have interfered with their sampling or coagulation status.  We have added a statement to the materials section stating how the sample size was chosen (see pg 3, line 99-101).  We can add a post hoc power calculation if the reviewer thinks that would be helpful.

Results: OK

Figures:

Comment 6:  It could just be how the upload occurred, but on my screen these figures are very small, and the text is not clear.

Response 6:  Thank you for pointing this out.  We have reconfigured the figures to make each panel (A, B, C) larger.  The text appears clear on our version and is hopefully improved for the reviewer (Fig 1 pg 5-6; Fig 2 pg 7-8)

Comment 7:  For your lettering is the a,b different non-pregnant mares? It is slightly confusing the way this is presented.

Response 7:  Thank you for pointing out the need for clarification here.  In Figure 1, the lowercase letters indicate significant difference over time in non-pregnant mares such that timepoints labeled with only a are different from those labeled with only b and that those labeled a, b are not different from any other timepoints.  Text has been added to the legend stating this (see pg 6, line 232-233)

Discussion:

Comment 8:  Line 230: While I do think your hypothesis is supported, in your limitations you list that the assay was not optimized completely, and your study is underpowered. I wonder if it would be fair to say in these initial findings and that more work would be needed to fully determine if there is a difference as seen in humans.

Response 8:  The authors agree and have modified the concluding statement and added a sentence indicating the need for a future, larger study to confirm the findings (see pg 8, line 256-260).

Comment 9:  Line 232: Do you think breed variation could have come into play for this? Your pregnant mare group was all AQHA where the non-group was very mixed. I think you do list this in your limitations but it may be useful to mention this here as well.

Response 9:  Thank you for this question.  We have added to this paragraph to clarify that differences over time and between the pregnant and non-pregnant groups may have contributed to the differences over time in the non-pregnant group.   We have included age and race as factors that have been associated with changes in fibrinolysis in humans and could also occur in horses (pg 8, line 264-268).  We also moved the paragraph discussing incomplete matching of the pregnant and non-pregnant mares up to directly follow this statement (pg 8-9, line 269-283)

Comment 10:  Line 250: Is it possible his was just not a high enough dose? I think you briefly mention this in your limitations but may want to expand on this a little bit more.

Response 10:  Thank you for this question.  We have added to the discussion of optimization of the assay (pg 9-10 line 323-351) as well as to the justification of the choice of the tPA doses (pg 9, line 297-301).

Reviewer 3 Report

Comments and Suggestions for Authors

This study investigates the fibrinolytic status of peripartum mares compared to non-pregnant mares using a modified thromboelastography (TEG) method incorporating tissue plasminogen activator (tPA). The authors observed relative hypofibrinolysis—characterized by stronger clots and reduced clot breakdown—in mares during late pregnancy and early postpartum, aligning with similar findings in human medicine. The study provides preliminary evidence that could inform improved management of hemorrhage in mares during the peripartum period. However, limitations such as small sample size, variability in assay performance, and unexpected trends in the control group temper the conclusions and underscore the need for further research.

Major Comments

  1. The small sample size (6–9 animals per group) significantly limits the statistical power and robustness of the conclusions. A larger cohort is necessary to validate findings.
  2. The tPA-modified TEG assay did not consistently induce the expected levels of fibrinolysis, indicating that assay conditions may not have been fully optimized for this species and study design.
  3. The use of frozen plasma rather than whole blood excludes the influence of cellular components on coagulation and fibrinolysis, which could alter the interpretation of results.
  4. Unexpected temporal variations in fibrinolysis parameters among non-pregnant mares complicate comparisons and raise concerns about underlying uncontrolled variables.
  5. The control and experimental groups differed in breed composition, which could confound results, especially since breed-related differences in hemostasis have not been fully investigated in horses.

Minor Comments:

  1. The manuscript would benefit from a clearer explanation of why certain tPA concentrations were selected, especially given the assay underperformance.
  2. More discussion of the implications of using mares from a university-owned herd (versus a broader population) would strengthen generalizability claims.
  3. Consider revising some repetitive or verbose sections in the introduction and discussion for better clarity and flow.
  4. Several typographical and formatting inconsistencies (e.g., duplicated punctuation, line breaks) should be addressed before final submission.
  5. The figures are informative but could benefit from clearer legends and axis labels to enhance standalone interpretability.
  6. The authors acknowledge the absence of additional coagulation and fibrinolysis markers, but a brief explanation of how these might have added value would be helpful.

Author Response

Comment 1:  This study investigates the fibrinolytic status of peripartum mares compared to non-pregnant mares using a modified thromboelastography (TEG) method incorporating tissue plasminogen activator (tPA). The authors observed relative hypofibrinolysis—characterized by stronger clots and reduced clot breakdown—in mares during late pregnancy and early postpartum, aligning with similar findings in human medicine. The study provides preliminary evidence that could inform improved management of hemorrhage in mares during the peripartum period. However, limitations such as small sample size, variability in assay performance, and unexpected trends in the control group temper the conclusions and underscore the need for further research.

 Response 1:  The authors thank you for your time and thoughtful review.  We have addressed your concerns to our best ability and believe the manuscript has been improved based on your suggestions.  We recognize the limitations of the study design and have added to our previous discussion of these issues including sample size and testing methodology to emphasize the need for cautious interpretation and further investigation.  Additionally, we have tempered our conclusions in the abstract and end of the manuscript.  New text is highlighted in yellow and text that has been moved to improve flow or clarity is highlighted in blue.

Major Comments

Comment 2:  The small sample size (6–9 animals per group) significantly limits the statistical power and robustness of the conclusions. A larger cohort is necessary to validate findings.

Response 2:  The authors recognize this limitation and have added text to the abstract and discussion indicating the need to repeat the study with a larger cohort to confirm the results in the abstract (pg 1, line 42-43), discussion (see pg 8, line 256-260), and conclusion (pg 10-11, line 375-377).  Further, we have added information explaining that the number samples was determined by available mares eligible for the groups in our research herd (pg 3, line 99-101).

Comment 3:  The tPA-modified TEG assay did not consistently induce the expected levels of fibrinolysis, indicating that assay conditions may not have been fully optimized for this species and study design.

Response 3: The authors recognize this limitation and have expanded on the discussion of optimization of the assay (pg 9-10, line 321-350)

Comment 4:  The use of frozen plasma rather than whole blood excludes the influence of cellular components on coagulation and fibrinolysis, which could alter the interpretation of results.

Response 4:  The authors recognize that there are potential benefits to performing the assay in whole blood which is stated in the discussion (pg 9, line 318-320) at the end of the paragraph where we justified the use of plasma samples for this study.  In designing the study we felt that the benefits of using plasma for future application, the successful performance of the assay in pooled plasma, and the use of a similar assay in human plasma made the choice to use plasma appropriate.  We have added reference to value of the human plasma assay to this paragraph (pg 9, line 311-313).

Comment 5:  Unexpected temporal variations in fibrinolysis parameters among non-pregnant mares complicate comparisons and raise concerns about underlying uncontrolled variables.

Response 5:  The authors recognize this limitation.  We have expanded/revised the paragraph in the discussion related to the changes over time identified in the non-pregnant mares.  We have moved the paragraph discussing the limitations of matching between the non-pregnant and pregnant mares to directly follow the sentence describing factors known to be associated with changes in fibrinolysis in humans and added a sentence recognizing that all of these factors could not be controlled for in this study. (pg 8-9, line 351-383)

We have separated the part of the original paragraph discussing the unexpected differences over time in the non-pregnant mares in an attempt to improve flow and emphasize the sentence stating that this presents a challenge to interpreting the differences between pregnant and non-pregnant mares. (pg 9, line 284-295).

Comment 6:  The control and experimental groups differed in breed composition, which could confound results, especially since breed-related differences in hemostasis have not been fully investigated in horses.

Response 6:  Thank you for this comment.  We have added race as a factor that has been associated with changes in fibrinolysis in humans (pg 8, line 267) and suggested that this might be similar to breeds in horses (pg 9, line 280).

Minor Comments:

Comment 7:  The manuscript would benefit from a clearer explanation of why certain tPA concentrations were selected, especially given the assay underperformance.

Response 7:  Thank you for this comment.  We have provided additional detail in the materials section (pg 3, line 121-124) and further explanation in the discussion (pg 9, line 297-301).

Comment 8:  More discussion of the implications of using mares from a university-owned herd (versus a broader population) would strengthen generalizability claims.

Response 8:  Thank you for this comment.  We have added a sentence describing management of the mares in the research herd indicating similarity to private breeding farms (pg 3, line 93-96). 

Comment 9:  Consider revising some repetitive or verbose sections in the introduction and discussion for better clarity and flow.

Response 9:  Thank you for this comment.  We have edited the introduction and discussion based on reviewer comments as well as flow of the manuscript.  We hope that we have made satisfactory changes and are happy to provide further revisions if requested with additional guidance from the reviewer.

Comment 10:  Several typographical and formatting inconsistencies (e.g., duplicated punctuation, line breaks) should be addressed before final submission.

Response 10:  Thank you for this comment.  We have reviewed the manuscript thoroughly and found several typographical errors and hope that the editor will find the current version acceptable.

Comment 11:  The figures are informative but could benefit from clearer legends and axis labels to enhance standalone interpretability.

Response 11:  Thank you for pointing this out.  We have reconfigured the figures to make each panel (A, B, C) larger.  The text appears clear on our version and is hopefully improved for the reviewer (Fig 1 pg 5-6; Fig 2 pg 7-8)

Comment 12:  The authors acknowledge the absence of additional coagulation and fibrinolysis markers, but a brief explanation of how these might have added value would be helpful.

Response 12:  Thank you for this comment.  A sentence has been added to the paragraph discussing this limitation (pg 10, line 352-354).

Round 2

Reviewer 3 Report

Comments and Suggestions for Authors

Accept that article.